# The impact of contact tracing and household bubbles on deconfinement strategies for COVID-19

Lander Willem [1✉], Steven Abrams [2,3], Pieter J. K. Libin[2,4,5], Pietro Coletti[2], Elise Kuylen[1,2], Oana Petrof[2], Signe Møgelmose[2,6], James Wambua [2], Sereina A. Herzog [1], Christel Faes[2], Philippe Beutels[1,7] & Niel Hens [1,2]

The COVID-19 pandemic caused many governments to impose policies restricting social interactions. A controlled and persistent release of lockdown measures covers many potential strategies and is subject to extensive scenario analyses. Here, we use an individual-based model (STRIDE) to simulate interactions between 11 million inhabitants of Belgium at different levels including extended household settings, i.e., "household bubbles". The burden of COVID-19 is impacted by both the intensity and frequency of physical contacts, and therefore, household bubbles have the potential to reduce hospital admissions by 90%. In addition, we find that it is crucial to complete contact tracing 4 days after symptom onset. Assumptions on the susceptibility of children affect the impact of school reopening, though we find that business and leisure-related social mixing patterns have more impact on COVID-19 associated disease burden. An optimal deployment of the mitigation policies under study require timely compliance to physical distancing, testing and self-isolation.

[1] Centre for Health Economic Research and Modelling Infectious Diseases, University of Antwerp, Antwerp, Belgium. [2] Data Science Institute, UHasselt, Hasselt, Belgium. [3] Global Health Institute, University of Antwerp, Antwerp, Belgium. [4] Artificial Intelligence Lab, Vrije Universiteit Brussel, Brussels, Belgium. [5] Rega Institute for Medical Research, Clinical and Epidemiological Virology, University of Leuven, Leuven, Belgium. [6] Centre for Population, Family and Health, University of Antwerp, Antwerp, Belgium. [7] School of Public Health and Community Medicine, The University of New South Wales, Sydney, NSW, Australia. ✉email: lander.willem@uantwerp.be

As the COVID-19 pandemic rose, there was an urgent need to understand the transmission dynamics and potential impact of COVID-19 on healthcare capacity and to translate these insights into policy. Mathematical modelling has been essential to inform decision-making by estimating the consequences of unmitigated spread in the initial phase, as well as the impact of non-pharmaceutical interventions. Gradually releasing society's lockdown while keeping the spread of the virus under control, requires detailed models to simulate the (non-) propagation of SARS-Cov-2. To this end, it is important to capture the heterogeneity in social encounters by accounting for a low number of intense contacts (e.g., between household members) and a high(er) number of more fleeting contacts (e.g., during leisure activities, commuting, or in shops)[1].

Transmission models at the level of the individual allow for flexibility to cope with chance, age and context, which is especially of interest to study exit strategies involving school, workplace, leisure activities and micro-scale policies[2,3]. Individual-based models (IBMs) pose a high burden on data-requirements, implementation and computation, however, the increasing availability of individual-level data facilitates thorough evaluation of specific intervention measures.

Understanding the interplay between human behaviour and infectious disease dynamics is key to improve modelling and control efforts[4]. Social contact data has become available for numerous countries[5,6] and has proven to be an invaluable source of information on the transmission of close contact infectious diseases[7,8]. Social contact patterns can be used as a proxy for transmission dynamics when relying on the "social contact hypothesis"[7]. Disease-related proportionality factors and timings enable the matching of age-specific mixing patterns with observed incidence, prevalence, generation interval and reproduction number. Social contact patterns in a transmission model can be adjusted to simulate behavioural change and assess possible intervention strategies[4].

Given the rising number of confirmed COVID-19 cases and hospital admissions in Belgium during the beginning of March 2020, all schools, universities, cultural activities, bars and restaurants were closed from March 14th onward. Additional measures were imposed on March 18th, with only work-related transport of essential workers allowed, and teleworking made the norm (termed "lockdown light" in comparison with stricter lockdowns in other countries). Hospital admissions peaked at the beginning of April, and declined afterwards[9]. Restrictive measures were gradually lifted from May 4th onward in terms of business-to-business (B2B), school, business-to-costumers (B2C) and leisure activities. There remains substantial uncertainty on the extent to which people complied with physical distancing guidelines during the deconfinement and how public awareness and interventions modified social contact characteristics. More specifically, did people mix in specific clusters and what was the effect of keeping distance, increased hygiene measures and wearing face masks? The nature of social contacts before and after the lockdown undoubtedly changed, and this affects the proportionality factors linking "contacts" with "transmission". Prior to the SARS-CoV-2 pandemic, simulation models for infectious diseases could rely on documented social contact behaviour as key input to model transmission dynamics. For COVID-19 predictions, there is however structural uncertainty on future social contact behaviour, implying that additional runs or improved parameter estimation would not reduce it. For example, the incremental effect when contact tracing is in place depends on the tendency of people to meet others. If the population stays put, the effect of contact tracing is minimal because it would be dominated by the effect of having only within-household mixing, and the epidemic would fade out. This structural uncertainty can be captured through different social mixing assumptions within each strategy assessment.

In what follows, we analyse the effect of repetitive leisure contacts in extended household settings (so called "household bubbles") on the transmission of SARS-Cov-2 and explore contact tracing strategies (CTS) with respect to coverage, sensitivity and timing. Our analyses are based on the open-source IBM "STRIDE", fitted to COVID-19 data from Belgium, with particular focus on transmission dynamics from adaptive social contact patterns.

## Results

We calibrated the transmission model up to April 30th, 2020, and continued all simulations up to August 31st to assess the impact of different deconfinement strategies. We start from a baseline scenario with step wise re-opening of B2B, schools and community activities including four assumptions capturing low and moderate increases in social mixing. Figure 1 presents the simulated hospital admissions over time from our baseline

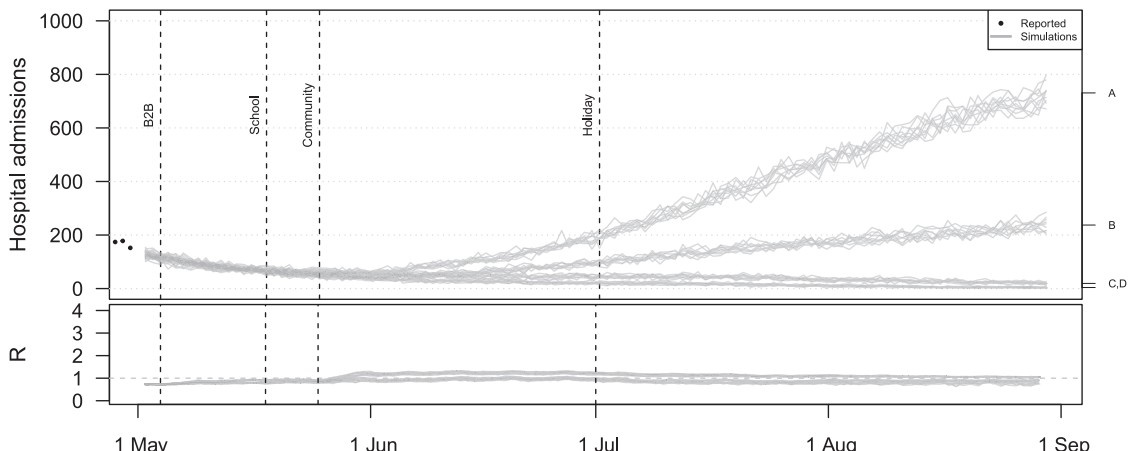

**Fig. 1 Hospital admissions and effective reproduction number (R) from the baseline scenario including four mixing assumptions.** All simulations include social restrictions from March 14th and the partial school reopening in May. For the B2B, the social mixing after the lockdown is assumed to double from the indicated point in time (marked on the right hand side with A and C) or to remain constant (B,D). Social mixing in the community is assumed to double (A,B) or to remain constant (C,D).

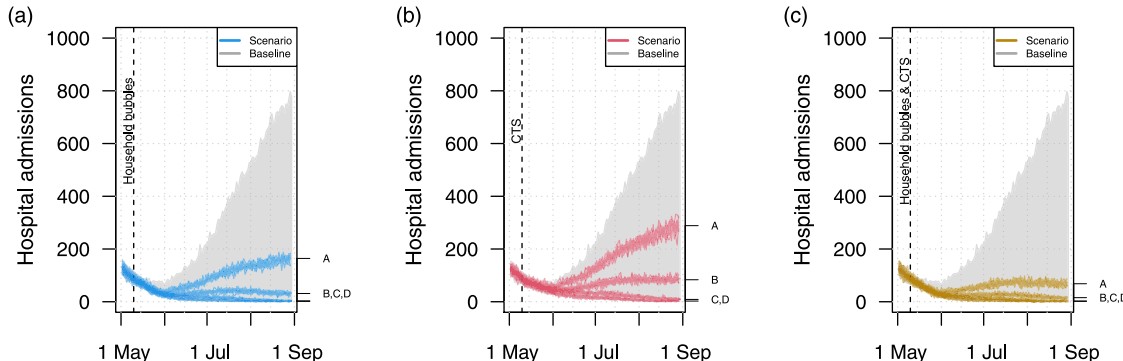

**Fig. 2 Impact of household bubbles and contact tracing on hospital admissions.** Hospital admissions over time when community mixing occurs in household bubbles (**a**), contact tracing strategy is in place (**b**), or both (**c**). All scenarios are based on the same natural disease history and quantitative mixing assumptions, but differ from the baseline in terms of the network structure and application of contact tracing from the given point in time. The mixing assumptions A,B,C,D are explained in the caption of Fig. 1. CST contact tracing strategy.

scenario with the timing of context-specific re-openings. Each grey line represents one stochastic trajectory of the simulator based on one social mixing assumption. The trajectories marked with A and B include an increase in community-related social mixing, which has a clear impact on the projected hospital admissions. The trajectories marked with A and C include an increase in B2B related mixing. Without an increase in community mixing, the effect of B2B seems minimal. We estimated the reproduction number before the lockdown to be 3.42 [3.41–3.49], which dropped below 1 during the lockdown. The reproduction number in our baseline scenario increases above 1 after the deconfinement for community contacts, which includes B2C and leisure activities.

Scenario analysis shows that social mixing in household bubbles, contact tracing and a combined strategy has a clear impact on the hospital admissions over time (Fig. 2). All scenarios are based on the same assumptions in terms of the absolute number of social contacts in line with the baseline scenario (Fig. 1). If people have fewer unique contacts, as in the scenario that considers household bubbles, the number of hospital admissions decreases. This is also the case with a strict follow up of symptomatic cases and their contacts when applying the contact tracing strategy (CTS). For both the household bubble and CTS scenario, the reduction is not sufficient if both B2B and community mixing doubles (trajectories marked with "A"), since the number of hospital admissions still increases over time. The combination of both strategies shows a stabilising effect for all social mixing assumptions under study.

Household bubbles are defined by connectivity, intergenerational mixing and size, which all have impact on our simulated hospital admissions, as presented by the projections for June and August in Fig. 3. Note that the reported distribution and summary statistics strongly depend on our mixing assumptions so they can only be used to show relative differences across scenarios. Our default scenario with household bubbles shows an average reduction in the average number of hospital admissions by 53% in June and by 75% in August as compared to the baseline scenario.

By not having leisure contacts outside the household bubble (i.e., if the household bubbles are fully connected 7 days a week), the average number of hospital admissions can be reduced by 93% by August. If household bubbles are less strict (i.e., fully connected 2 days a week), the effect is less pronounced but the average number of hospitalisations in August can still be 41% less compared to our baseline. If household bubbles consist of households of which the ages of the oldest household members

can differ up to 20 or 60 years and multiple generations are allowed within one household bubble, the effectiveness of this strategy decreases. The reduction in daily hospital admission by June is only 43% if 60-year differences are allowed. By August, after 2 months of school holiday, there are almost no differences between the different age gap scenarios, which suggests that the limited exposure of children in our simulations might not exploit the full extent of the intergenerational mixing within household bubbles. If household bubbles consist of three households, they almost replace all community contacts in our simulations, which results in fewer hospital admissions in the long run due to the closed network topology. If people mix within household bubbles of size 4, the average number of leisure contacts increases compared to our baseline mixing assumptions, which explains the reduced effectiveness of the household bubble approach for June. However, due to the closed nature of these extended bubbles and restricted number of unique contacts, the average number of hospital admissions by August is comparable to situation with household bubbles of size 2, despite the increased contact rates.

**Contact tracing strategy (CTS).** The follow up of symptomatic cases by strict isolation and contact tracing, i.e., screen their contacts with isolation if infected, shows a substantial effect on the average number of hospital admissions (Figs. 2 and 3). We project an average reduction in hospital admissions of 22% in June and 57% in August with the CTS in place, assuming that 70% of the symptomatic cases are subjected to contact tracing and comply with home isolation. The combination of contact tracing and repetitive social mixing in household bubbles has the potential to reduce the average number of hospital admissions up to 87% by August. This approaches the effect of strict household bubbles, but clearly allows more freedom in terms of social mixing.

Our CTS results are based on different assumptions with respect to timing and success rates of tracing, testing, and compliance to home isolation if infected. We performed a sensitivity analysis to challenge our CTS assumptions that 70% of the symptomatic cases are considered as index case, 10% false-negative tests, a 90% success rate for tracing household members and 70% for other contacts. The false-negative predictive value of testing, due to the sampling, lab-testing and assessment of the treating physician, is important but we still observed an impact of the CTS with 30% false-negative tests if the coverage is high enough (see Fig. 4). By varying the success rate of contact tracing per index case, the relative number of hospital admissions ranged from 35% to 60% of the base case scenario without CTS. Tracing

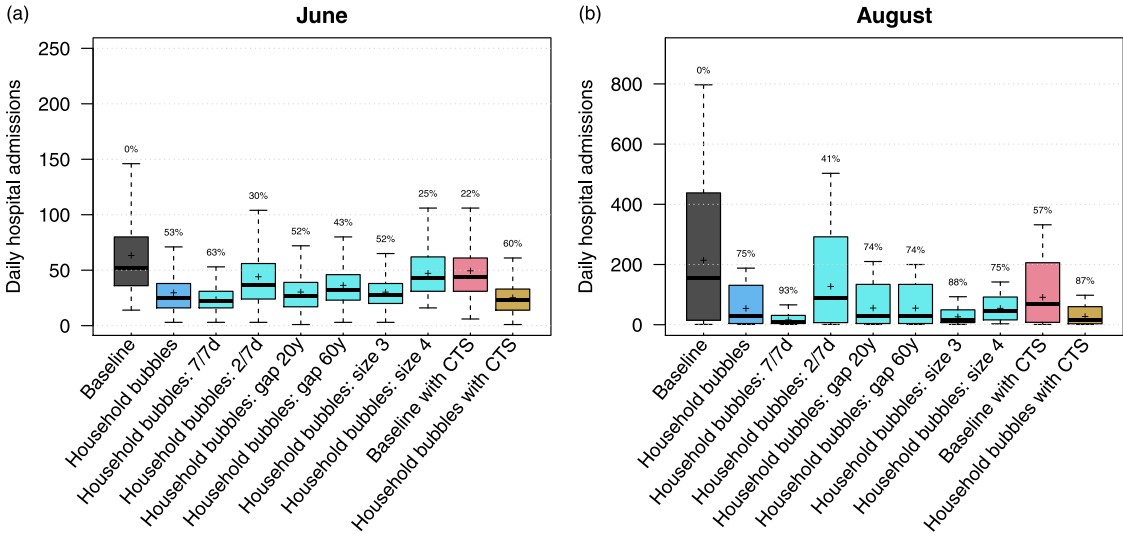

**Fig. 3 Daily hospital admissions per scenario.** Distribution of the daily hospital admissions by June (**a**) and August (**b**) per scenario. The results are presented as the median (line), quartiles (box), 2.5 and 97.5 percentiles (whiskers) and average (cross) of 40 model realisations (i.e., ten stochastic runs for each of the four social contact behaviour assumptions). The percentage on top of the whiskers indicates relative reduction of the scenario average with respect to the baseline. CTS contact tracing strategy.

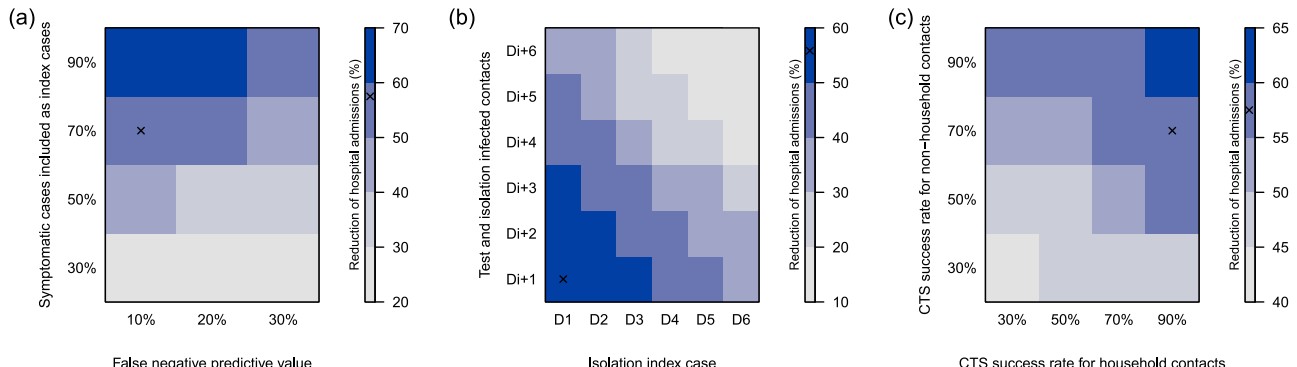

**Fig. 4 Reduction of hospital admissions due to contact tracing according to the symptomatic cases included as an index case, the false-negative predictive value of testing, delays and the success rate of tracing, testing and isolating household and (non-)household contacts.** Timings are expressed relative to symptom onset of the index case (D0), and days after testing the index case (e.g., Di + 1). All simulations start from the baseline scenario and assume a 50% and 30% reduction of B2B and community contacts, respectively, compared to pre-lockdown observations. The "x" marks the default settings, which are used if a parameter is not shown. CST contact tracing strategy.

non-household contacts seems to have the most impact, since their absolute number can be higher compared to household contacts. However, tracing and testing household contacts, which are easier to define and accessible via the index case, also makes a difference. With a maximum delay between symptom onset of the index case and isolation of infected contacts up to 4 days, we observed the best results in terms of averted hospital admissions. If this delay increases, the efficiency of CTS drops. Note that if index cases are identified and isolated only 6 days after symptom onset, the tracing should be very fast or there will not be much left to gain.

**Location-specific re-opening.** We analysed the effect of location-specific deconfinement strategies on the total number of hospital admissions between May and August with two assumptions on the susceptibility for children up to 17 years of age: equally susceptible or only half as susceptible compared to adults (+18 years)[10]. The results are presented in Fig. 5. Starting from the baseline and each time leaving one location-specific re-opening out, we observed most impact of community mixing for both susceptibility-related assumptions. Without an increase in

community mixing, the simulated hospital admissions decrease by almost 70%. Without an increase in B2B-related social mixing, the total number of hospital admissions between May and August is still 50% less compared to our baseline scenario. The effect of household bubbles and CTS on the cumulative hospital cases is similar for both susceptibility assumptions. As expected, the impact of school re-opening is strongly associated with the assumption on age-specific susceptibility. Assuming that children are equally susceptible compared to adults, we observe an increase in hospital admissions up to 96% and 181% compared to our baseline scenario if schools reopen up to primary or secondary education, respectively. Re-opening primary schools without any precautionary measure such as smaller class groups, class separation, masks, and increased hand hygiene seems worse than opening all schools with a 50% reduction of transmission. The re-opening of pre-schools has limited effect on the simulated hospital admissions according to the mixing assumptions under study. Assuming an age-specific susceptibility, re-opening primary schools has less impact on the predicted number of hospital admissions. If all children up to 17 years of age go back to school with precautionary measures, we observed an increase of hospital

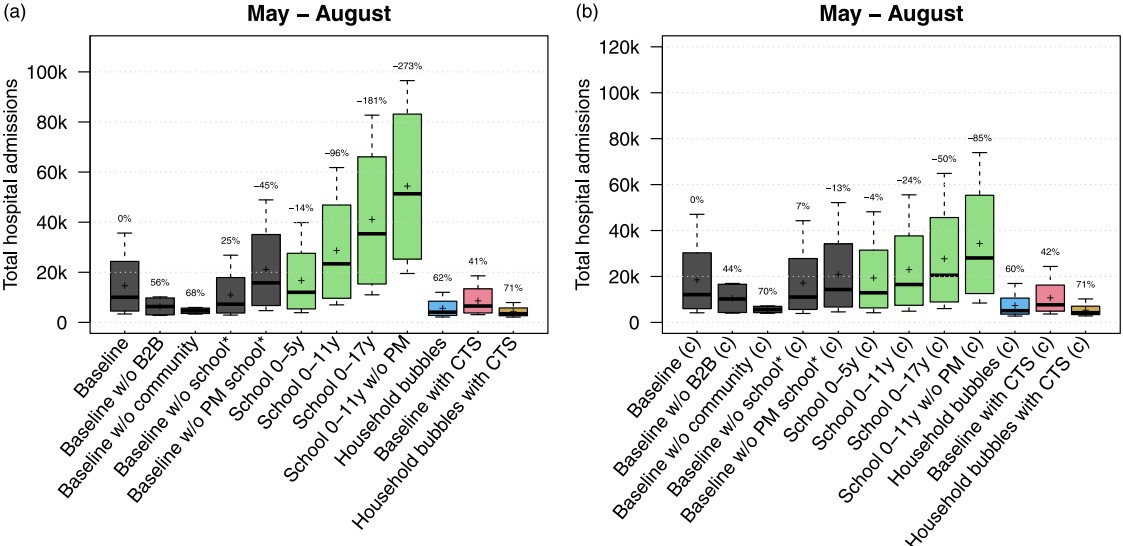

**Fig. 5 Impact of location-specific re-openings and age-specific susceptibility.** Total hospital admissions per scenario from May to August assuming that children between 0 and 17 years are equally susceptible as adults (**a**) or only half as susceptible compared to adults (**b**). The results are presented as the median (line), quartiles (box), 2.5 and 97.5 percentiles (whiskers) and average (cross) of 40 model realisations (i.e., ten stochastic runs for each of the four social contact behaviour assumptions). The percentages on top of the whiskers indicate the average reduction in hospital admissions with respect to the baseline. CTS contact tracing strategy, w/o without, PM precautionary measures at school.

cases up to 50% relative to our baseline scenario. Note that these scenarios do not take contact tracing or other physical distancing measures into account but express the transmission potential at school. Combining different scenarios to define the required contact tracing efficiency or other measures to enable schools to reopen is subject of future research.

**Sensitivity and robustness analyses.** Given the correlated nature of our model parameters, different combinations can give a similar fit for the first wave but might lead to different outcomes for the deconfinement strategies in the scenario analyses. To assess the robustness of our results, we simulated the main scenarios with an ensemble of model parameter sets. The resulting projections in terms of hospital admissions over time (Supplementary Fig. 20) show more variation but the average reduction in hospital admissions (Supplementary Fig. 21) does not change.

We performed a robustness analysis on the number of stochastic realisations for the main scenarios and observe more spikes in the hospital admissions over time with an increasing number of stochastic realisations (Supplementary Figs. 16 and 17) but no differences in the average hospital admissions for June and August (Supplementary Fig. 18). Details are provided in the Supplementary Information.

## Discussion

Uncertainty on social mixing after a lockdown plays a crucial role in predicting the outcome of deconfinement strategies. How will people behave socially if restrictions are relieved? A deconfinement strategy can allow for economic or leisure activities, but people might still limit their contacts or they might fully exploit the renewed freedom, beyond what is requested but hard to regulate or enforce. In addition to contact frequency, contact intensity (duration, intimacy, indoor/outside location, etc.) also plays a role in the transmission dynamics. To handle this structural uncertainty in our simulations, we included different social mixing assumptions as part of assessing deconfinement strategies. Our baseline strategy, including four different mixing assumptions, is not chosen to capture the observed situation as much as

possible, but to analyse the relative impact of mutually exclusive scenarios as in comparative effectiveness research.

Parallel modelling work for the UK[11] showed that social bubbles reduced cases and fatalities by 17% compared to an unclustered increase of contacts. Social bubbles may be very effective if targeted towards small isolated households with the greatest need for additional social interactions and support. Their analyses confirm that social bubble strategy is an effective way to expand contacts while limiting the risk of a resurgence of cases.

We found a great potential for CTS to reduce transmission and hospital admissions, but it might not be enough to control future waves. The simulations in Fig. 2 with the least strict physical distancing still show an increase in hospital admissions with CTS in place. Only if the number of contacts is limited and/or contacts take place in closed networks such as the household bubbles, the CTS is sufficient to keep the hospital admissions low. The relative proportion of symptomatic cases that is included in the CTS as index case is driving the efficiency. Also, timing is of the essence and contact tracing should start at the latest 4 days after symptom onset of the index case. The short serial interval makes it difficult to trace contacts due to the rapid turnover of case generations[12]. Keeling et al.[1] concluded that rapid and effective contact tracing can be highly effective in the early control of COVID-19, but places substantial demands on the local public-health authorities. We did not include or analyse the enhancing/spiralling effect when infected contacts are subsequently included as an index case. This could be one way to improve case finding to include as index case, which we implicitly incorporated in our strategy. Another effect of this spiralling approach might be a reduction of the workload given overlapping contacts with a previous index case. However, the timing of physical contacts and testing might interfere with this optimisation procedure. We did not look into this but focus on the basic principles and stress the potential of CTS.

Kucharski et al.[13] also reported on the effectiveness of physical distancing, testing, and a CTS for COVID-19 in the UK. They concluded that the combination of a CTS with moderate physical distancing measures is likely to achieve control. They also used an

IBM with location-specific mixing and transmission parameters and similar natural history of the disease. Their model is different in the number of contacts, which is fixed to 4, and social contact pools for school, work and other, are defined at a lower degree of granularity compared to our model. We are able to identify the class members and direct colleagues of infected individuals, and can confirm their conclusions on CTS and isolation strategies. We both stress the potential of CTS but warn that additional physical distancing measures are required to be successful.

Kretzschmar et al.[14] computed effective reproduction numbers with CTS and social distancing in place by considering various scenarios for isolation of index cases and tracing and quarantine of their contacts. Without a delay in testing and tracing and with full compliance, the effective reproduction number was reduced by 50%. With a testing delay of 4 days, even the most efficient CTS could not reach effective reproduction numbers below 1. We did not express the impact of CTS on the reproduction number, though also found a tipping point in the CTS effectiveness if contact tracing starts 4–5 days after symptom onset of the index case. To improve the early detection of cases, the use of universal testing in which the entire population is screened on a regular basis is promising[15].

School closure is considered a key intervention for epidemics of respiratory infections due to children's higher contact rates[16,17], but the impact of school closure depends on the role of children in transmission. Davies et al.[10] conclude that interventions aimed at children might have a relatively small impact on reducing SARS-CoV-2 transmission, particularly if the transmissibility of subclinical infections is low. This is also the conclusion from our scenario analyses where we assume children (<18 years) to be half as susceptible as adults (+18 years).

Other IBM applications have been reported[3,18–20] to simulate combinations of non-pharmaceutical interventions by targeting transmission in different settings, such as school closures and work-from-home policies. Modelling the isolation of cases in safe facilities away from susceptible family members or by quarantining all family members to prevent transmission has shown substantial impact. Models that explicitly include location-specific mixing are very relevant for studying the effectiveness of non-pharmaceutical interventions, as these are more dependent on community structure than, e.g., with vaccination[18]. However, implementing the available evidence into a performant and tailor-made model that addresses a wide range of questions about a variety of strategies is challenging[2,19].

Although our analysis is applied to Belgium, our findings have wider applicability. We considered the effect of universal adjustments in terms of social mixing (isolation, repetitive contacts, contact tracing). We modelled 11 million unique inhabitants with detailed social contact patterns by age and location. Hence, we can compare model results with absolute incidence numbers in the absence of premature herd immunity effects due to a reduced population size. The latter can be an issue for models that use a scaling factor to obtain final results. Our individual-based model provides a high-resolution, mechanistic explanation of the reproduction number and transmission dynamics that are relevant on a global scale.

Any model is a simplification of reality and, therefore, depends on the assumptions made. In addition, our spatially explicit IBM is calibrated on national hospitalisation data so uncertainty is inevitably underestimated. As such, we rely on scenario analyses and further sensitivity analyses are necessary. Model results should therefore be interpreted with great caution. Our IBM is a mechanistic mathematical model that uses conversational contacts as a proxy of events during which transmission can occur. By definition, SARS-CoV-2 infection events that occurred through the environment (e.g.,

contaminated surfaces) are covered by these conversational contacts.

The use of antiviral drugs in combination with CTS can reduce the effect of local outbreaks[21]. This kind of pharmaceutical intervention is not incorporated in the current analysis. We focused on the transmission dynamics in the general population and did not consider care homes separately in our analysis. They form predominantly a sink for infections, with high morbidity and mortality, but are not likely to drive the transmission. To focus on the disease burden in the elderly, the social interactions within elderly homes and with their environment become more important[22]. We did not include aspects related to travel or weather conditions (UV light, humidity, temperature), which may impact both transmission and social contact behaviour in ways that are still largely unknown.

## Methods

**Model structure.** This work builds on a stochastic individual-based model (IBM) we developed for influenza[23,24] and measles[25]. Our model is representative for the population of Belgium, covering 11 million unique individuals, runs in discrete time steps of 1 day while accounting for adjusted social contact patterns during weekdays, weekends, holiday periods, illness and the influence of public awareness and imposed policy measures. More details on the model structure, population, social contact patterns and stochastic realisations are provided in the Supplementary Information.

**Disease natural history.** The health states in the IBM follow the conventional stages of susceptible, exposed, infectious and recovered, with the infectious health state divided in pre-symptomatic, symptomatic and asymptomatic. For every infected individual, we sample the onset and duration of each stage based on the distributions in Supplementary Table 2.

**Social contact patterns.** Social contact patterns for healthy, pre- and asymptomatic individuals are parameterised by a diary-based study performed in Belgium in 2010–2011[26–28]. Contact rates at school and at work are conditional on school enrolment and employment, respectively. We account for behavioural changes of symptomatic cases using observations made during the 2009 H1N1 influenza pandemic in the UK[29], by reducing presence at school and work with 90%. Based on the same study, we reduce community engagement with 75% when experiencing symptoms. Transmission-relevant contact behaviour within the household is assumed not to change when a household member develops symptoms.

**Parameter estimation.** We estimated transmission and lockdown characteristics based on reported hospital admissions[9], initial doubling time (i.e., before the lockdown)[30] and serial sero-prevalence data[31] up to May 1st. Afterwards, multiple restrictive measures in Belgium were relaxed, which is the focus of our scenario analysis. Details on the model parameters and our multi-criteria iterative procedure are provided in the Supplementary Information. Our iterative estimation procedure resulted in an ensemble of parameter sets that match our three reference criteria. From this ensemble, we selected a single best parameter set based on the average log-likelihood function value to match the observed hospital admissions over time, since this is the model outcome of main interest. The per-case average number of secondary cases in a susceptible population, which corresponds to the basic reproduction number $R_0$, was estimated to be 3.42, which is in line with estimates from a meta-analysis[32] and other modelling studies for Belgium[33,34]. Within our final model parameter ensemble, the reproduction number ranged between (3.41 and 3.49). The transmission model starts with 263 (236–307) infected cases on February 17th. The hospital probability for symptomatic cases over 80 years is 40% (35%–46%). From March 14th onward, the social contacts related to B2B decreased linearly to 14% (7%–30%) over 7 (5–7) days. Contacts in the community during lockdown decreased to 15% (13%–18%) of pre-lockdown contact levels after 7 (5–7) days.

**Household bubbles.** We defined a "household bubble" as a unique combination of two households in which the oldest household members cannot differ >3 years in age and are linked via their community contacts during weekends. The age-specific component is included to reduce intergenerational mixing, which is subject of sensitivity analyses with age-differences of 20 and 60 years. The assignment of household bubbles in STRIDE proceeds in a random order and if no matching household is available, the household is not assigned to any household bubble. This procedure enables us to assign > 95% of the population to a household bubble. These bubbles are exclusive and remain fixed throughout the simulation from May 11th onward.

We assume households in a social bubble to be fully connected 4 days out of 7 (i.e., the contact probability between any two bubble members per day is 4/7 =

0.57). We also test a higher and lower level of connectivity in terms of 7/7 and 2/7 days per week, respectively. Social contacts in a household bubble are implemented as a substitute of leisure contacts in the community and can be seen as repetitive leisure contacts with the same individuals. Therefore, the community contacts are reduced in proportion to the household bubble mixing to keep the overall contact rate unchanged. We also test household bubbles consisting of 3 and 4 households, where the number of household bubble contacts exceeds the number of simulated community contacts in our scenarios, so the total number of contacts increased. Symptomatic individuals have no social contacts with members of other households within their household bubble.

**Contact tracing strategy (CTS).** We implement CTS to assess their impact on hospital admissions if 70% of the symptomatic cases are considered to be index cases. Each index case is placed in home isolation 1 day after symptom onset. One day later, unique contacts are traced and tested at a success rate of 90% for household members, and 70% for non-household members. We assume a false-negative predictive value of 10%, as a combined outcome of sampling, lab-testing, and clinical assessment of the treating physician. We performed sensitivity analyses regarding the proportion of symptomatic cases included as an index case, the false-negative predictive value, the success rate to reach (non-)household contacts and contact tracing delays. The effect of these CTS parameters is tested using one of the social mixing assumptions (as described in the next paragraph).

**Scenario analyses.** We defined different strategies by location-specific decon-finement strategies with structural uncertainty about social contact behaviour after a lockdown. As such, we incorporated four mixing assumptions in our baseline scenario to capture a low and moderate increase in social contacts related to business-to-business (B2B) and community activities. By modelling reductions in social mixing, we implicitly assume people either make fewer contacts compared to the pre-pandemic situation or the contacts they make are less likely to lead to transmission. For example, some transmission will be prevented by more frequent hand washing, distancing or the use of masks[35]. If we assume that social mixing at workplaces increased from 25% to 50%, we estimate the impact of "what if the risk of acquiring infection at work doubles compared to during lockdown, but remains still 50% less compared to pre-pandemic times". Table 1 presents the social mixing details for each scenario.

In our baseline scenario, we accounted for an increase of B2B mixing (i.e., contacts while at work) from May 4th up to 50% of the pre-pandemic observations. Business-to-consumer (B2C) and leisure transmission is harder to single out using social contact data within our model structure. To model the relaunch of economic activities and other (leisure) activities in the community, we incorporated a limited increase of community mixing up to 30% in our scenario analyses starting from May 25th. Note that we do not claim that the increase of community mixing is estimated to be 30% or restricted to this level, but we provide insights up to 30%.

For schools, we assumed a 50% reduction of transmission due to precautionary measures (smaller class groups, class separation, increased hand hygiene, etc.) and performed sensitivity analyses to explore the effect of these measures. We aligned

**Table 1 Scenario definitions.**

| Scenario | Description |
|---|---|
| Baseline | During the lockdown, the social contacts related to B2B and in the community reduced to 15% of observed behaviour prior the lockdown. Schools are closed and household mixing did not change. For the deconfinement phase, we consider four social mixing assumptions based on B2B and community-related activities. The social contacts related to B2B increase to 25% or 50% of the pre-pandemic mixing patterns from May 4th. Community mixing is assumed to remain 15% or increase up to 30% from May 25th onward. All runs contain an age-specific partial school reopening from May 18th according the Belgian regulations at that time (see Supplementary Table 7) and assuming a 50% reduction of transmission at school due to precautionary measures. |
| Baseline w/o B2B. | Baseline scenario without increase in B2B mixing (=fixed to 25%). |
| Baseline w/o community | Baseline scenario without increase in community mixing (=fixed to 15%). |
| Baseline w/o school* | Baseline scenario without partial school reopening. |
| Baseline w/o PM at school* | Baseline scenario without precautionary measures at schools. |
| School 0–5 years | Baseline scenario with reopening of pre-schools with precautionary measures. |
| School 0–11 years | Baseline scenario with reopening of pre- and primary schools with precautionary measures. |
| School 0–17 years | Baseline scenario with reopening of pre-, primary and secondary schools with precautionary. measures |
| School 0–11 years w/o PM | Baseline scenario with reopening of pre- and primary schools without precautionary measures. |
| Household bubbles | Community mixing is partially replaced by social contacts within household bubbles consisting of two households of which the oldest two household members are part of the same weekend community and their ages can differ by up to 3 years. The mixing intensity equals the equivalent of being fully connected 4 days per week. Interaction within the household bubbles is possible from May 11th onward. |
| Household bubbles: 7/7 days | Household bubble scenario in which all members are in contact every day (connected 7 days per week), hence almost no community contacts remain. |
| Household bubbles: 2/7 days | Household bubble scenario in which the members are less connected, the equivalent of a visit 2 days per week, hence more community contacts remain. |
| Household bubbles: size 3 | Household bubble scenario with three households per bubble. The equivalent of being fully connected 4/7 days a week equals the assumed number of community contacts in our baseline scenario. As such, most leisure contacts are within the household bubble. |
| Household bubbles: size 4 | Household bubble scenario with four households per bubble and all leisure contacts are within the household bubble. The size of the household bubble surpasses the number of leisure contacts in the baseline scenario, hence the absolute number of contacts increases in this scenario. |
| Household bubbles: age gap 20 years | Household bubble scenario in which the ages of the two oldest household members can differ by up to 20 years. |
| Household bubbles: age gap 60 years | Household bubble scenario in which the ages of the two oldest household members can differ by up to 60 years. |
| Baseline with CTS | Baseline scenario with contact tracing strategy starting on May 11th in which 70% of the symptomatic cases are included, 90% of the household contacts and 70% of the non-household contacts are successfully traced, tested and isolated if infected. The false-negative predictive value is 10%. |
| Household bubbles and CTS | Household bubble scenario with contact tracing as specified above. |
| (child) or (c) | Scenarios including age-specific susceptibility in which children (0–17 years) are only half as susceptible compared to adults (+18 years). During the lockdown, the social contacts related to B2B and in the community reduced to 24% and 14%, respectively, of observed behaviour prior the lockdown. |

All reductions in social mixing are relative to observed social contact patterns before the lockdown. B2B: business-to-business, CTS: contact tracing strategy, w/o: without, PM: precautionary measures.
*B2B* business-to-business, *CTS* contact tracing strategy, *w/o* without, *PM* precautionary measures.

the baseline scenario with the school regulations and timings for Belgium (see Supplementary Table 7). In addition, we also included more general scenarios for re-opening pre-, primary and secondary schools from May 18th onward to make our analysis more explorative. We model that all schools close on July 1st, in line with the start of the national summer holiday period (until August 31st).

The Belgian government further relaxed restrictions in May 2020 by allowing additional contacts within the household context. We adopted a strict approach using household bubbles of two households of a similar generation based on the age of the oldest household member. To align a combined approach of household bubbles and contact tracing, both strategies start in our simulations on May 11th. We did not include additional region-specific distancing measures.

**Age-specific susceptibility**. To fully explore age-specific effects, especially for school-related scenarios, we additionally calibrated our transmission model assuming that children (0–17 years) are only half as susceptible compared to adults (+18 years)[10]. The methods are provided in Supplementary Information.

**Sensitivity and robustness analyses**. During the parameter estimation, we identified an ensemble of parameter sets at the intersection of the best scoring model runs according to the observed hospital admissions, doubling time before the lockdown and serial sero-prevalence. The results presented in the main text are based on the single best parameter set, but we repeated the main scenarios (baseline, household bubbles, CTS and the combination of both) with the final ensemble of model parameters. To validate our choice for presenting results based on 10 stochastic realisations, we also ran our main scenarios with 20, 40 and 80 stochastic realisations.

**Reporting summary**. Further information on research design is available in the Nature Research Reporting Summary linked to this article.

## Data availability

The series of COVID-19 cases and hospital admissions are publicly available from the Belgian Institute of Public Health (Sciensano) at https://epistat.wiv-isp.be/covid/covid-19.html. The synthetic population data we used for Belgium are made available on ZENODO[36].

## Code availability

We provide all code and model configuration scripts in a public archive of our open-source GitHub repository[37].

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

## Acknowledgements

The authors are very grateful for access to the data from the Belgian Scientific Institute for Public Health, Sciensano, and from the Vaccine & Infectious Disease Institute (VaxInfectio), University of Antwerp. We thank several researchers from the SIMID COVID-19 consortium from the University of Antwerp and Hasselt University for numerous constructive discussions and meetings. L.W., S.A., P.J.K.L. and N.H. gratefully acknowledge support from the Research Foundation Flanders (FWO) (postdoctoral fellowships 1234620N and 1242021N, and RESTORE project G0G2920N). This work also received funding from the European Research Council (ERC) under the European Union's Horizon 2020 research and innovation programme (P.C., S.A.H. and N.H., grant number 682540—TransMID project; C.F., P.B. and N.H. grant number 101003688—EpiPose project). P.B. and N.H. acknowledge funding from the Antwerp Study Centre for Infectious Diseases (ASCID) and the Methusalem-Centre of Excellence consortium VAX-IDEA. We used computational resources and services provided by the Flemish Supercomputer Centre (VSC), funded by the FWO and the Flemish Government, with special thanks to the CalcUA-team (FB and SB). The funders had no role in study design, data collection and analysis, decision to publish, or preparation of the manuscript.

## Author contributions

L.W., P.B. and N.H. conceived the study. S.A., P.J.K.L., P.C., O.P., S.M., J.W., S.A.H. and C.F. contributed to the data collection and analysis. P.J.K.L. and E.K. contributed to the software development. All authors contributed to the final version of the paper and approved the final manuscript.

## Competing interests

The authors declare no competing interests.
