## [Peer Review File · Nature Communications]

Reviewer #1 (Remarks to the Author):

Review of The impact of contact tracing and household bubbles on deconfinement strategies for COVID-19: an individual-based modelling study by L. Willem et al.

Summary: The authors address a very important and timely subject using an individual based model (IBM) with detailed heterogeneous contact structure, which takes into account different contact patterns for different age groups, and differential susceptibility and infectivity by age group. In particular, this study is very timely because it addresses the potential impacts of reopening schools and businesses, and compares contact tracing scenarios and the use of household bubbles as a means of control. Including the age-based heterogeneous contact structure is a key factor to consider within this context. The authors use rich data sets for Belgium to parameterize their IBM. Number of hospital admissions is used as the metric for comparison for different contact tracing scenarios and testing accuracies.

Comments: The manuscript and supplemental material are well-structured and well-written. I list a few questions/concerns below.

1. The abstract can be construed as stating that school closures may have little impact on the COVID-19 burden, however, the results don't seem to support this conclusion. The authors compare two cases: (1) children are equally susceptible to the virus as adults, and (2) susceptibility is age-dependent, and increases with age. In the case where susceptibility is age-independent, the authors predict an increase in hospital admissions of 126-295% when schools re-open, and if susceptibility is age-dependent, this increase is predicted to be up to 122%. These seem like substantial increases, and the abstract should more accurately reflect that, particularly since there is increasing evidence now that children are better vectors of transmission than previously thought.

2. I would like to see more detail about the baseline mechanistic transmission model. The model has an SEIR framework, which the authors clearly note, however, it is not clear to me how they included the pre-symptomatic period. Table S2 summarizes the distributions for parameters related to the duration of the incubation period, symptomatic period, infectious period, and pre-symptomatic period, however, these periods are not all distinct in reality, but instead overlap. I would like to see the assumptions behind the timeline of disease progression clearly stated.

3. The authors use uniform distributions for the incubation, symptomatic, infectious period, and pre-symptomatic infectious period. The referenced manuscript ([15] in the supplement) states that a gamma distribution was fit to serial interval data and the incubation period was assumed to be lognormal. The authors should explain their choice of uniform and how they chose the ranges for their uniform distributions.

4. In Table S2, the mean and confidence intervals for 3 metrics (generation interval, doubling 1 time, and R_0) are estimated from 10 realizations of the IBM. This doesn't seem like enough realizations given the stochasticity and uncertainty in model parameters. Much simpler stochastic models exhibit huge fluctuations from one realization to the next, often requiring closer to 1000 realizations to appropriately estimate the mean and variance.

5. In the supplement (S3), the authors describe how they estimate the age-specific proportion symptomatic (see eqn (2) and Figure S5), but do not seem to incorporate any uncertainty in these estimates. The authors should address why they chose these values to be fixed in the model.

6. Related to the previous comment, on p.4 of the manuscript, the authors state that sum of squared residuals was used to estimate symptomatic proportions from seroprevalence data (i.e. ordinary least squares fitting). This assumes the data are Gaussian – is this a reasonable assumption? Something like approximate Bayesian computation (ABC) could potentially be used to estimate posterior distributions for the parameters, and therefore better estimates of parameter uncertainty, which can then be fed into the IBM.

7. In supplement S3, the authors state that they assume an overall proportion of symptomatic cases in the population of 50%, and then distribute those cases by age group. What is the potential consequence of fixing this proportion rather than letting it change dynamically over time?

8. In supplement S4, the authors describe how they used reported hospital admissions to estimate the timing of the first COVID-19 introduction into Belgium, the number of initial cases, and the transmission probability. I would like the authors to clarify the following sentences from the first paragraph of this section: “We performed a grid search using the sum of squared residuals of the reported... To assess the stochastic robustness of the parametric setting, we calculated the average score of 10 stochastic realisations for each parameter set.” I would also like the authors to comment on their choice of the sum of squared residuals, versus a likelihood model more suited to count data. Was any uncertainty in these parameters included in the model simulations?

9. Under the Results section, I was surprised to read that increasing the age gap between households from 3 to 60 had limited impact on transmission potential. I would expect, particularly under the school reopening scenario, that this would increase contacts between high risk individuals over 65 and school-aged children, and therefore increase hospital admissions. Do the authors have some intuition for why the impact would be limited?

10. Minor comment: I think in the phrase, “enables us to assign $\pm 95\%$ of the population to a household bubble” in the second-to-last paragraph on p. 3, this should simply be +95%.

Reviewer #2 (Remarks to the Author):

The authors have done an analysis of potential COVID-19 interventions, simulating social "bubbles," contact tracing, and by adapting a model called STRIDE, which they had previously developed to model influenza in Belgium. The model has a great deal of fidelity to the current demographics of Belgium: it constructs synthetic households that closely resemble actual Belgian households in household size and demographic composition. They have developed algorithms to simulate complex social mixing patterns. They fit the model to data describing cases and seroprevalence using non-linear least squares, and unsurprisingly, the fits they get appear to be very good.

I would be much more positive about the study if I could understand why they consider their model and analysis to be valid. If the model has been fit to all of the data available (i.e. the black dots on Figure 1), then it is not surprising that the fits look so good, but why should I believe the results of the simulated effect sizes of contact tracing and bubbles in STRIDE? The authors have thoroughly discussed the model structure, but I can't find where they discuss what parameters changed in their simulations to mimic the change in policy that occurred around April 1 through the present day. I think I understand how they fit the model to the epidemic phase (e.g. late Feb and early March), but what did they change in the simulation to make it fit the declines so well after April 1? This would be a very important conclusion on its own. What data did they use to model that change? How do the scenarios they evaluate represent a change compared with the last 10 weeks or so? This would seem to be very critical point for a reader to understand before they could know how to evaluate the rest of the work.

Even though the authors say, "Social mixing patterns represent a key uncertainty in COVID-19 prediction models and is therefore central to our analysis," I am struggling to understand the difference between two things: 1) how they can explain what has happened so far through changes in social mixing patterns; and 2) what recommendations they are making for the country as it resumes normal activity.

I don't feel I can evaluate the merits of the paper based on the material I have in front of me, so I will reject it. I'm very sorry to do it, because it otherwise seems like a very good study.

Point-by-point reply on the reviews of “The impact of contact tracing and household bubbles on deconfinement strategies for COVID-19: an individual-based modelling study” (NCOMMS-20-28269-T)

Reviewer #1

Summary: The authors address a very important and timely subject using an individual based model (IBM) with detailed heterogeneous contact structure, which takes into account different contact patterns for different age groups, and differential susceptibility and infectivity by age group. In particular, this study is very timely because it addresses the potential impacts of reopening schools and businesses, and compares contact tracing scenarios and the use of household bubbles as a means of control. Including the age-based heterogeneous contact structure is a key factor to consider within this context. The authors use rich data sets for Belgium to parameterize their IBM. Number of hospital admissions is used as the metric for comparison for different contact tracing scenarios and testing accuracies.

Comments: The manuscript and supplemental material are well-structured and well-written. I list a few questions/concerns below.

We thank the reviewer for the appreciation of our work and address the comments inline.

1. The abstract can be construed as stating that school closures may have little impact on the COVID-19 burden, however, the results don't seem to support this conclusion. The authors compare two cases: (1) children are equally susceptible to the virus as adults, and (2) susceptibility is age-dependent, and increases with age. In the case where susceptibility is age-independent, the authors predict an increase in hospital admissions of 126-295% when schools re-open, and if susceptibility is age-dependent, this increase is predicted to be up to 122%. These seem like substantial increases, and the abstract should more accurately reflect that, particularly since there is increasing evidence now that children are better vectors of transmission than previously thought.

We thank the reviewer for this comment and we agree that this statement should be presented in a more nuanced way. We explored the effect of school reopening for two assumptions on the susceptibility for children and explain the results for both assumptions. Since there is no clear evidence on one direction yet, we changed the sentence in the abstract to:

The susceptibility of children affects the impact of a (partial) reopening of schools, though we found that social mixing patterns related to business and leisure activities are driving the COVID-19 burden.

2. I would like to see more detail about the baseline mechanistic transmission model. The model has an SEIR framework, which the authors clearly note, however, it is not clear to me how they included the pre-symptomatic period. Table S2 summarizes the distributions for parameters related to the duration of the incubation period, symptomatic period, infectious period, and pre-symptomatic period, however, these periods are not all distinct in reality, but instead overlap. I would like to see the assumptions behind the timeline of disease progression clearly stated.

We agree with the reviewer that this description could be improved. In order to clarify this, we devoted a separate section (S4) in the Supplementary Material to the natural history of the disease as included in the model. Figures S6 and S7 show the state transitions and dynamics with (pre-)symptomatic and asymptomatic stages.

3. The authors use uniform distributions for the incubation, symptomatic, infectious period, and pre-symptomatic infectious period. The referenced manuscript ([15] in the supplement) states that a gamma distribution was fit to serial interval data and the incubation period was assumed to be lognormal. The authors should explain their choice of uniform and how they chose the ranges for their uniform distributions.

The ranges were based on the literature, but the rationale for using uniform distributions was indeed not clear. During the revision, we revised the distributions based on the literature and provided all details and figures in Section S4 of the Supplementary Information entitled: "Natural disease history". One important remark is that the transmission model runs in discrete time steps of one day, hence we discretized the distributions of the transition parameters.

4. In Table S2, the mean and confidence intervals for 3 metrics (generation interval, doubling time, and R_0) are estimated from 10 realizations of the IBM. This doesn't seem like enough realizations given the stochasticity and uncertainty in model parameters. Much simpler stochastic models exhibit huge fluctuations from one realization to the next, often requiring closer to 1000 realizations to appropriately estimate the mean and variance.

The reported interval for R_0 was limited because all simulations were based on 1 parameter set for the transmission and lockdown parameters. For the revision, we repeated the parameter estimation procedure (see our reply on Question 6) and report the doubling time and R_0 based on runs with different model parameters in Table S4 and the ranges clearly increased. We omitted the generation interval to prevent the comparison of the individual-based modelling approach for this transmission characteristic with compartmental modelling estimates.

To validate our choice for presenting results based on 10 stochastic realisations, we also ran our main scenarios (baseline, household bubbles and contact-tracing) with 20, 40 and 80 stochastic iterations. We observed stochastic differences in the hospital admissions over time, but the results in terms of the average number of hospital admissions did not alter. We also ran simulations using the ensemble of estimated model parameters and observed again stochastic differences in the hospital admissions over time, but stable results in terms of the average number of hospital admissions. The sensitivity analyses on the stochastic realisations and the ensemble approach are described in the Supplementary Material S10 and S11 and are summarized in the main text.

5. In the supplement (S3), the authors describe how they estimate the age-specific proportion symptomatic (see eqn (2) and Figure S5), but do not seem to incorporate any uncertainty in these estimates. The authors should address why they chose these values to be fixed in the model.

The age-specific proportion of symptomatic cases is indeed based on the results presented in Figure 2b from Wu et al (2020). Their results contain merely uncertainty for age groups over 60 years of age, and in the absence of data to define an appropriate extent of uncertainty in the form of a distribution associated with these parameters, we opted to fix these parameters to have values equal to the aforementioned estimates. Note that these proportions are incorporated as probabilities to specify disease transitions on the individual-level, hence we do account for stochasticity with respect to the development of symptoms.

6. Related to the previous comment, on p.4 of the manuscript, the authors state that sum of squared residuals was used to estimate symptomatic proportions from seroprevalence data (i.e. ordinary least squares fitting). This assumes the data are Gaussian – is this a reasonable assumption? Something like approximate Bayesian computation (ABC) could potentially be used to estimate posterior distributions for the parameters, and therefore better estimates of parameter uncertainty, which can then be fed into the IBM.

We thank the reviewer for the critical note about our approach of using the sum of squared residuals. To improve our methodology, we switched our objective function to the Poisson log-likelihood function as a statistic to be minimized in order to assess how well the model described the observed data given a set of parameter values. Given our interest in hospital admissions, incidence and transmission dynamics, we added the doubling time in March as reference data in addition to hospital admissions and serial sero-prevalence. To select optimal parameter combinations, we used the intersection of the best scoring model runs for each criterion and repeated this incremental process four times. Details are provided in the Supplementary Material S7.

We acknowledge that our parameter estimation procedure was not sufficiently explained in the initial submission and a grid-search using discrete parameter values might be suboptimal. Therefore, we switched within our iterative procedure to a Latin Hypercube Design consisting of 1000 parameter sets containing a value for 7 model inputs. We accounted for parameter uncertainty in the sensitivity analysis by using an ensemble of parameter sets we derived from the fourth iteration. The estimation of posterior distributions for the model parameters using, for example, ABC, would be interesting to compare our approach with. The transmission and lockdown parameter we obtain are in line with the findings reported by Abrams et al (2020) using MCMC. Our estimation approach does not provide the granularity from a Bayesian approach, but fits our purpose to capture average transmission and lockdown characteristics to evaluate deconfinement scenarios in a realistic context.

7. In supplement S3, the authors state that they assume an overall proportion of symptomatic cases in the population of 50%, and then distribute those cases by age group. What is the potential consequence of fixing this proportion rather than letting it change dynamically over time?

We estimated an age-specific proportion of symptomatic cases based on the age-specific relative susceptibility to symptomatic infection from Wu et al (2020) and a weighted overall proportion before the lockdown of 50% based on Li et al (2020). The proportion of symptomatic cases per age group is handled as a disease related feature and fixed over time. The proportion of symptomatic cases in the population depends on the age of the newly infected cases, which is driven by social contact and transmission dynamics. Therefore, given the temporal aspects of social contact behaviour and restrictions, the overall proportion of symptomatic cases in the population can change over time.

We added the explanation above in Section S3 of the Supplementary Material.

8. In supplement S4, the authors describe how they used reported hospital admissions to estimate the timing of the first COVID-19 introduction into Belgium, the number of initial cases, and the transmission probability. I would like the authors to clarify the following sentences from the first paragraph of this section: “We performed a grid search using the sum of squared residuals of the reported... To assess the stochastic robustness of the parametric setting, we calculated the average score of 10 stochastic realisations for each parameter set.” I would also like the authors to comment on their choice of the sum of squared residuals, versus a likelihood model more suited to count data. Was any uncertainty in these parameters included in the model simulations?

As answered to Question 6, we switched to the negative Poisson log-likelihood function to assess how well the model described the observed data, given the nature of our reference data. This improved our iterative parameter estimation procedure with multiple stochastic realisations. We did not include overdispersion in the likelihood model to prevent additional complexity and to increase the computational burden, and since our goal is to capture average transmission and lockdown dynamics for a mechanistic transmission model and not the estimation of posterior distributions for the model parameters.

Parameter uncertainty is now captured via the ensemble-based sensitivity analyses, which we discussed in our answer on Question 4.

9. Under the Results section, I was surprised to read that increasing the age gap between households from 3 to 60 had limited impact on transmission potential. I would expect, particularly under the school reopening scenario, that this would increase contacts between high risk individuals over 65 and school-aged children, and therefore increase hospital admissions. Do the authors have some intuition for why the impact would be limited?

Our conclusions on the age gap between households were merely driven by the predicted number of daily hospital admissions in August. Looking at the predictions for June (also in Figure 2), the difference between an age gap of 60 years is more pronounced. The effectiveness of the household bubble in June with a limited age gap of 3 years (53% reduction

compared to baseline) is reduced if 60-year age gaps are allowed (43% reduction). We thank the referee for this critical reflection and elaborated more on this in the Results section:

If household bubbles consist of households of which the ages of the oldest household members can differ up to 20 or 60 years and multiple generations are allowed within one household bubble, the effectiveness of this strategy decreases. The reduction in daily hospital admission by June is only 43% if 60-year differences are allowed. By August, after 2 months of school holiday, there are almost no differences between the different age gap scenarios, which suggests that the limited exposure of children in our simulations might not exploit the full extent of the intergenerational mixing within household bubbles.

10. Minor comment: I think in the phrase, “enables us to assign $\pm 95\%$ of the population to a household bubble” in the second-to-last paragraph on p. 3, this should simply be +95%.

Adapted, thank you.

Reviewer #2

The authors have done an analysis of potential COVID-19 interventions, simulating social "bubbles," contact tracing, and by adapting a model called STRIDE, which they had previously developed to model influenza in Belgium. The model has a great deal of fidelity to the current demographics of Belgium: it constructs synthetic households that closely resemble actual Belgian households in household size and demographic composition. They have developed algorithms to simulate complex social mixing patterns. They fit the model to data describing cases and seroprevalence using non-linear least squares, and unsurprisingly, the fits they get appear to be very good.

I would be much more positive about the study if I could understand why they consider their model and analysis to be valid. If the model has been fit to all of the data available (i.e. the black dots on Figure 1), then it is not surprising that the fits look so good, but why should I believe the results of the simulated effect sizes of contact tracing and bubbles in STRIDE? The authors have thoroughly discussed the model structure, but I can't find where they discuss what parameters changed in their simulations to mimic the change in policy that occurred around April 1 through the present day. I think I understand how they fit the model to the epidemic phase (e.g. late Feb and early March), but what did they change in the simulation to make it fit the declines so well after April 1? This would be a very important conclusion on its own. What data did they use to model that change? How do the scenarios they evaluate represent a change compared with the last 10 weeks or so? This would seem to be very critical point for a reader to understand before they could know how to evaluate the rest of the work.

We thank the reviewer for these fundamental questions on our modelling study and the way we present our results. Initially, the model was used to perform short-term forecasts on deconfinement strategies to inform the Belgian government on the application of (household) bubbles and location specific exit strategies. In contrast, this manuscript is not intended to discuss predictions or prediction methods, but to provide insights into transmission dynamics of SARS-Cov-2 before, during and after a lockdown. The goal of this paper is to describe the impact of different deconfinement strategies for (future) lockdowns while accounting for different sources of uncertainty. To do so, we make use of a mechanistic transmission model on the level of the individual that is based on census and survey data to inform demography and social contact behaviour. Additionally, we included the natural history of SARS-Cov-2 based on the literature. The population, contact and disease layers emerge into the population behaviour, which we used to calibrate the transmission potential within the model to the Belgian situation and to capture the effect of the lockdown measures. The model calibration, i.e., the parameter estimation, is performed with observed data up to April 2020. The scenario analysis targets the period May - August 2020 in which we only vary social contact assumptions and report the emerging burden of disease in each simulation. In the methods section, we elaborated on the parameter estimation procedure and explicitly included the parameters that we estimated and for which period they apply:

We estimated transmission and lockdown characteristics based on reported hospital admissions [9], initial doubling time (i.e., before the lockdown) [17] and serial seroprevalence data [18] up to May 1st. Afterwards, multiple restrictive measures in Belgium were relaxed, which is the focus of our scenario analysis. Details on the model parameters and our multi-criteria iterative procedure are provided in the

Supplementary Material. Our iterative estimation procedure resulted in an ensemble of parameter sets that match our three reference criteria. From this ensemble, we selected a single best parameter set based on the average log-likelihood function value to match the observed hospital admissions over time, since this is the model outcome of main interest. The per-case average number of secondary cases in a susceptible population, which corresponds to the basic reproduction number R_0 , was estimated to be 3.42, which is in line with estimates from a meta-analysis [19] and other modelling studies for Belgium [20, 21]. Within our final model parameter ensemble, the reproduction number ranged between (3.41–3.49). The transmission model starts with 263 (236–307) infected cases on February 17th. The hospital probability for symptomatic cases over 80 years is 40% (35%–46%). From March 14th onward, the social contacts related to B2B decreased linearly to 14% (7%–30%) over 7 (5–7) days. Contacts in the community during lockdown decreased to 15% (13%–18%) of pre-lockdown contact levels after 7 (5–7) days.

From May 2020 onward, the Belgian government allowed several relaxations on the social restrictions, which is the focus of this study, so we stopped the model fitting there. I.e. all model parameters except the social mixing after April 2020 were kept constant in the scenario analyses. The goal of our study is to translate non-pharmaceutical interventions in terms of adjusted social contact behaviour into the averted disease burden. We adapted Figure 1 and 2 to make clear from which point in time, we do not use observed data anymore but focus on the gradually re-opening of business-to-business, business-to-costumers, schools, community contacts and assessed the impact according to underlying contact dynamics with open or closed networks (= household bubbles) and contact tracing.

The “prediction uncertainty” we referred to in our initial submission is replaced by “structural uncertainty” on the social contact behaviour, for two reasons. Firstly, to stress that our results are no predictions and the goal is not to express which scenario is most likely. Secondly, to emphasise the structural nature of this uncertainty, implying additional runs, or improved parameter estimation would not reduce it. The fundamental question is whether people will increase their social mixing, and how, once formal restrictions are relieved. The effect of, for example, a contact tracing strategy largely depends on the tendency of people to meet others. In a scenario without substantial disease transmission, the effect of contact tracing is negligible. We elaborate more on this in the introduction:

Restrictive measures were gradually lifted from May 4th onward in terms of business-to-business (B2B), school, business-to-costumers (B2C) and leisure activities. There remains substantial uncertainty on the extent to which people complied with physical distancing guidelines during the deconfinement and how public awareness and interventions modified social contacts characteristics. More specifically, did people mix in specific clusters and what was the effect of keeping distance, increased hygiene measures and wearing face masks? The nature of social contacts before and after the lockdown undoubtedly changed, and this affects the proportionality factors linking “contacts” with “transmission”. Prior to the SARS-CoV-2 pandemic, simulation models for infectious diseases could rely on documented social contact behavior as key input to model transmission dynamics. For COVID-19 predictions, there is however structural uncertainty on future social contact behavior, implying that additional runs or improved parameter estimation would not reduce it. For example, the incremental

effect when contact tracing is in place depends on the tendency of people to meet others. If the population stays put, the effect of contact tracing is minimal because it would be dominated by the effect of having only within-household mixing, and the epidemic would fade out. This structural uncertainty can be captured through different social mixing assumptions within each strategy assessment.

In the initial submission, we shortly discussed why our baseline scenario did not follow the reported trends and why the scenarios with household bubbles or contact tracing performed better. We removed this part to prevent confusion about our scenarios being predictions, and added the following:

Our baseline strategy including the 4 different mixing assumptions is not chosen to capture the observed situation as much as possible, but to analyse the relative impact of mutually exclusive scenarios as in comparative effectiveness research.

Even though the authors say, "Social mixing patterns represent a key uncertainty in COVID-19 prediction models and is therefore central to our analysis," I am struggling to understand the difference between two things: 1) how they can explain what has happened so far through changes in social mixing patterns; and 2) what recommendations they are making for the country as it resumes normal activity.

The sentence "Social mixing patterns represent a key uncertainty in COVID-19 prediction models and is therefore central to our analysis," gave too much weight to the role of social mixing patterns and is rephrased in the revised manuscript. Firstly, we elaborate more in the Introduction on the role of social mixing as a proxy for transmission dynamics and the structural uncertainty on social contact behaviour for COVID-19 predictions in the introduction.

Understanding the interplay between human behavior and infectious disease dynamics is key to improve modelling and control efforts [4]. Social contact data has become available for numerous countries [5, 6] and has proven to be an invaluable source of information on the transmission of close contact infectious diseases [7, 8]. Social contact patterns can be used as a proxy for transmission dynamics when relying on the "social contact hypothesis" [7]. Disease-related proportionality factors and timings enable matching age- specific mixing patterns with observed incidence, prevalence, generation interval and reproduction number. Social contact patterns in a transmission model can be adjusted to simulate behavioural change and assess possible intervention strategies [4].

Secondly, we advocate the use of social contact patterns in transmission modelling. Our individual-based transmission model is based on demography, social interactions and the natural disease history of SARS-Cov-2. By changing one aspect in such a mechanistic model, the population-level burden of disease changes. In contrast to demography and the natural disease history, social contact behaviour is prone to rapid change due to personal decisions or forced measures. With our model, we show that by estimating the reduction in social contacts within the transmission framework, we were able to capture the first wave of COVID-19 related hospital admissions in Belgium. We rephrased our modelling focus to:

In what follows, we analyse the effect of repetitive leisure contacts in extended household settings (so called “household bubbles”) on the transmission of COVID-19 and explore contact tracing strategies with respect to coverage, sensitivity and timing. Our analyses are based on the open-source IBM “STRIDE”, fitted to COVID-19 data from Belgium, with particular focus on transmission dynamics from adaptive social contact patterns.

Our recommendations are that closed networks in household bubbles, or other network concepts, can make a substantial difference in terms of transmission and consequently burden of disease. People can maintain numerous social contacts as long as their network is closed. The nature of leisure contacts is more open compared to household, school and work contacts, which is a liability in terms of transmission. In addition to resume normal activity, contact tracing and isolation is key but this should be complete within 4 days after symptom onset of the index case and reach a high coverage. If contact tracing is not optimally implemented, additional risk management in terms of reduced or directed social contact behaviour is still required. This is summarised in the abstract

Conclusions. Next to the absolute number and intensity of physical contacts, also their repetitiveness impacts the transmission dynamics and COVID-19 burden. The combination of closed networks and contact tracing seems essential for a controlled and persistent release of lockdown measures, but requires timely compliance to the bubble concept, testing, reporting and self-isolation.

I don't feel I can evaluate the merits of the paper based on the material I have in front of me, so I will reject it. I'm very sorry to do it, because it otherwise seems like a very good study.

We are thankful for the honest feedback of the reviewer on our study and we hope that we were able to convince the reviewer of the merits of our revised manuscript, both based on the included changes and the point-by-point replies.

Reviewer #1 (Remarks to the Author):

The authors have significantly improved the presentation of the model details and parameter estimation procedure in the supplement. I appreciate that they repeated their parameter estimation with an improved likelihood model more suitable for their count data. I am pleased that they compared the means across simulations for increasing numbers of simulations (20, 40, 80) and were able to show consistency in the computed means. Perhaps this is a consequence of count data on the order of 100s. I like the Latin Hypercube sampling approach to generate the ensemble of parameters; five realizations for each parameter set still strikes me as low, but I assume this is a computational limitation. Below I list some minor corrections. Otherwise, I feel that the authors have made substantial efforts to appropriately address my concerns.

Grammatical errors/typos in the following lines of supplemental materials:

- 605: “one day prior **to** symptom”
- 610: “as prior”
- 649 and 728: equation for $P_{\text{transmission}}$ does not follow from previous equation.
- 668: “required hygiene **and** physical”
- 676: “is **the** subject”
- 705-706: “incremented **by** 2%”
- 711: rephrase sentence “The first iteration...”
- 723: halve->half
- 763: “include more variation **in** the hospital admissions”